# Montmorillonite Catalyzed Synthesis of Novel Steroid Dimers

**DOI:** 10.3390/molecules28207068

**Published:** 2023-10-13

**Authors:** Aneta M. Tomkiel, Adam D. Majewski, Leszek Siergiejczyk, Jacek W. Morzycki

**Affiliations:** 1Faculty of Chemistry, University of Bialystok, Ciołkowskiego 1K, 15-245 Bialystok, Poland; a.majewski@uwb.edu.pl (A.D.M.); nmrbial@uwb.edu.pl (L.S.); 2Doctoral School of Exact and Natural Sciences, University of Bialystok, Ciołkowskiego 1K, 15-245 Bialystok, Poland

**Keywords:** steroid dimers, 1,4-phenylene-linked dimers, montmorillonite, DHEA, cholesterol, steroids

## Abstract

The reactions of sterols (androst-5-en-3β-ol-17-one, diosgenin, and cholesterol) and their tosylates with hydroquinone aimed at the synthesis of *O*,*O*-1,4-phenylene-linked steroid dimers were studied. The reaction course strongly depended on the conditions used. The study has shown that the major reaction products are the elimination products and unusual steroid dimers resulting from the nucleophilic attack of the hydroquinone C2 carbon atom on the steroid C3 position, followed by an intramolecular addition to the C5–C6 double bond. A different reaction course was observed when montmorillonite K10 was used as a catalyst. The reaction of androst-5-en-3β-ol-17-one under the promotion of this catalyst afforded the *O*,*O*-1,4-phenylene-linked steroid dimer in addition to the disteroidal ether. The formation of the latter compound was suppressed by using 3-tosylate as a substrate instead of the free sterol. The reactions of androst-5-en-3β-ol-17-one tosylate and cholesteryl tosylate with hydroquinone catalyzed by montmorillonite K10 carried out under optimized conditions afforded the desired dimers in 31% and 67% yield, respectively.

## 1. Introduction

During the last few decades steroid dimers have emerged as an interesting new area of steroid chemistry. The first review article on steroid dimers was published by Li and Dias in 1997 [1]. Since then, numerous reports have appeared on these compounds, which have been reviewed in the book [2] and in several review articles by Nahar et al. [3,4]. In the beginning, steroid dimers, which were formed as byproducts of certain reactions, were considered mere curiosities. Later, steroid dimers were found in natural sources, e.g., japindine isolated from the root bark of *Chonemorpha macrophylla* [5], crellastatins from marine sponge *Crella* sp. [6], cephalostatins from tiny marine worm *Cephalodiscus gilchristi* [7], and ritterazines from tunicate *Ritterella tokioka* (Figure 1) [8]. The highly cytotoxic cephalostatins and ritterazines inspired chemists to attempt their chemical synthesis [9,10,11]. The synthesis of these dimeric steroidal pyrazine alkaloids and their analogs was particularly important because the natural sources of these compounds are extremely limited. These intensive studies were covered by several review articles [12,13,14,15]. Oligomeric steroids with or without spacer groups can be used as chiral building blocks to construct artificial receptors and as architectural components in biomimetic/molecular recognition chemistry. Davis has briefly reviewed his work in this area, directed toward the construction of enzyme mimics [16]. The study on dimerization and oligomerization of the cholic acid skeleton as an architectural building block was particularly intensive [17,18,19]. Various methods of synthesis of steroid dimers with linear [20] or cyclic [21] structures, e.g., Sonogashira coupling, Yamaguchi esterification, Wurtz reaction, ring closing metathesis, etc., were employed [22]. In the last decades, the interesting chemical, biological, and physical properties of the growing number of steroid dimers synthesized or isolated from living organisms have triggered increased activity in this field. The large number of steroid dimers described so far include compounds that have shown properties as catalysts [23], artificial receptors [24], molecular umbrellas [25], as well as activity as sulfatase inhibitors [26], antimalarials [27], and cytotoxic and antiproliferative agents [28,29], amongst others. Recent solid-state studies revealed that several crystalline steroid dimers act as molecular rotors [30].
Figure 1Naturally occurring steroid dimers.
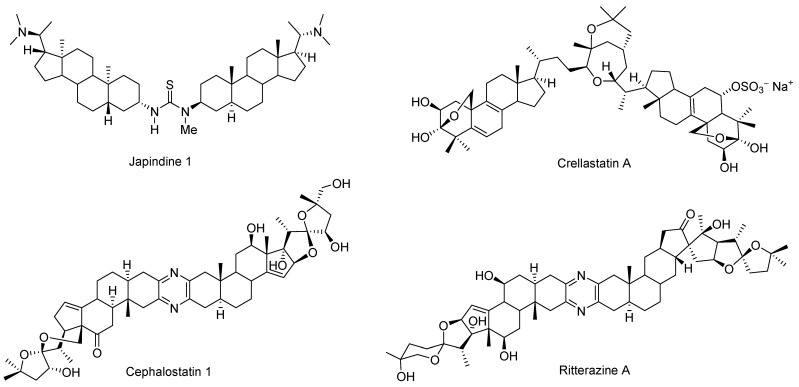

Figure 2Target steroid dimers (**1**–**3**) and starting materials (**4**–**6**).
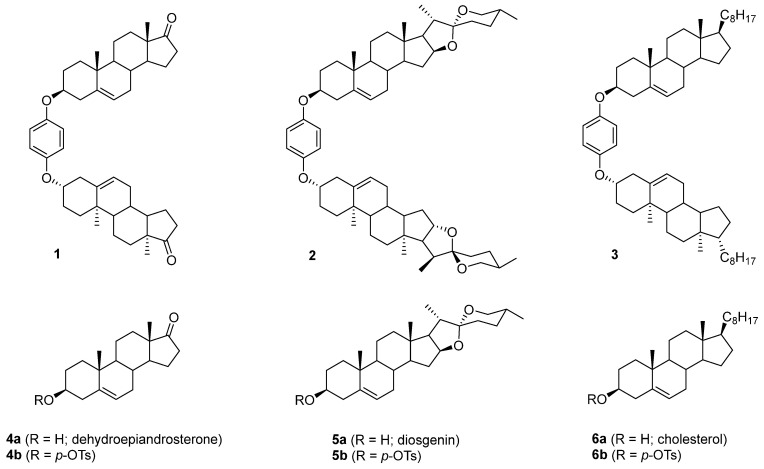


The aim of this research was the synthesis of steroid dimers, in which steroid units are connected by an *O*,*O*-1,4-phenylene linker (Figure 2). These compounds (**1**−**3**) were selected as potential molecular rotors. Molecular motors are an important class of nanomolar-sized molecular machines that use different energy sources to generate unidirectional mechanical motion. In the crystal lattice, they contain both a static part and a fragment capable of free rotation [31]. Compound **1** has a carbonyl group at C17, which allows modification of the structure by functional group manipulation. Compounds **1** and **2** have not been described before, while compound **3** was previously obtained in low yield from cholesterol hydroquinone ether as an undesired product of an electrochemical reaction [32]. In the present study, a chemical procedure was elaborated for the efficient synthesis of dimers **1**−**3** from readily available sterols, androst-5-en-3β-ol-17-one **4a** (dehydroepiandrosterone; DHEA), (25*R*)-spirost-5-en-3β-ol **5a** (diosgenin), and cholesterol **6a**, respectively.

## 2. Results and Discussion

The synthetic plan of dimers **1**–**3** assumed the conversion of starting sterols **4a**–**6a** into the corresponding *p*-tosylates **4b**–**6b** and their reaction with hydroquinone. It is well known that upon leaving *p*-TsO^−^ the mesomerically stabilized homoallylic carbocation (Figure 1) is generated, which can react with a nucleophilic reagent, e.g., hydroquinone, in the 3β position. For stereoelectronic reasons, only 3β-substituted steroid products can be formed. In fact, the nucleophile may also attack the 6β position, but the resulting product (so-called *i*-steroid) is thermodynamically less stable and is not formed as a product under acidic conditions. However, the reaction of androst-5-en-3β-ol-17-one tosylate **4b** (2 equiv.) with hydroquinone (1 equiv.) did not afford the expected dimer **1**. There was no reaction in boiling acetone (even in the presence of *p*-TsOH), while the analogous reaction carried out in dioxane (24 h at reflux) yielded mostly the nonpolar elimination products **7** and **8** (Table 1; run 1). The fast solvent−free reaction at 120 °C (run 2) gave a mixture of several products, which were carefully analyzed.

In addition to a mixture of dienes **7** and **8**, androst-5-en-3β-ol-17-one hydroquinone ether **9** and unexpected products **10** and **11** as stereomeric mixtures were obtained. The compound **9** could be an intermediate in the synthesis of dimer **1,** but the desired compound **1** was not found among the reaction products. Instead, the different dimers were isolated, **11a**–**11c**, and their likely precursors, **10a** and **10b**. The similar reaction products, but in a different ratio (more elimination products), were formed in xylenes at reflux for 3 h (run 3). The structures of the products prove that both nucleophilic sites of hydroquinone, the oxygen atom and the carbon atom C2 of the aromatic ring, take part in the reaction. Interestingly, during the reaction, the hydroquinone carbon atom binds to the steroid C3 position, while the oxygen atom binds to C5. The direction of the reaction was deduced from the analysis of ^1^H NMR spectra. The chemical shifts of protons at C3 were shifted upfield (compared to δ 4.34 in tosylate **4b**) and appeared at δ below 3 ppm. Therefore, the oxygen atom in this compound cannot be attached to C3. It seems that compounds **10b** and **11b** resulted from an attack of the carbon atom of hydroquinone on the mesomeric carbocation, which is formed upon the leaving of *p*-TsO^−^ from **4b** (Figure 2).
molecules-28-07068-t001_Table 1Table 1The results of tosylate **4b** (2 equiv.) reaction with hydroquinone (1 equiv.) under different conditions.Run No./Reaction ConditionsStructure of ProductsConversion
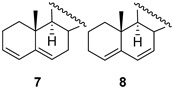

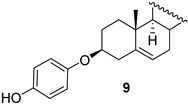

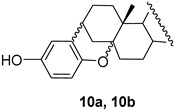

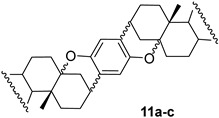
1: Dioxane, reflux, 24 h16%<5%--30%2: 120 °C, 5 min23%12%**10a**: <5%, **10b**: <5%**11a**: 7%, **11b**: 5%, **11c**: 9%100%3: Xylenes, reflux, 3 h48%13%**10a**: <1%, **10b**: <1%**11a**: <5%, **11b**: 5%, **11c**: 5%100%4: Ball mill, 48 h7%19%**10a**: <1%, **10b**: <1%**11a**: <5%, **11b**: <5%, **11c**: 5%50%5: Acetone, MW, 110 °C, 6 h47%8%**10a**: <1%, **10b**: <1%**11a**: <1%, **11b**: <5%, **11c**: 5%90%6: Dioxane, US, 45 °C, 12 h5%---15%

This attack is possible from the steroid β side only due to stereoelectronic reasons. In the next step, an intramolecular addition of phenol to the steroid C5–C6 double bond occurred, leading to a six-membered ring formation. Compound **10b** obtained this way may undergo the same reaction sequence to provide the dimer **11b**. The mechanism of the dimer **11a** formation (via **10a**) is less obvious (Figure 3). Apparently, the attack of hydroquinone on steroid tosylate **4b** occurred from the less hindered α side in this case. However, the reaction timing is not clear. It could be either a nucleophilic substitution of 3-tosylate with the carbon C2 of hydroquinone followed by an intramolecular addition to the C5–C6 double bond or a concerted attack of hydroquinone on both 3-tosylate and the double bond.

The dimers **11a**, **11b,** and **11c** were separated by column chromatography since they differ in polarity. The Rf values of dimers determined by their migration on TLC plates three times developed in the solvent system hexane—ethyl acetate (74:26) were 0.77, 0.66, and 0.71, respectively. The C2 symmetrical dimers **11a** and **11b** were isolated in similar amounts, while the mixed unsymmetrical dimer **11c** was obtained in larger quantities. Optimized structures of dimers **11a** and **11b** are presented in Figure 3. The ^1^H NMR spectra of **11a** and **11b** were very similar, except for protons in the A-ring region. Particularly, the chemical shifts of protons at C3 and C3′ in these dimers proved different. The two-proton signal in dimer **11a** came out at δ 2.90 ppm, while an analogous two-proton signal in **11b** appeared at δ 2.85 ppm. Of course, the ^1^H NMR spectrum of mixed dimer **11c**, which consists of two different steroid units, showed both proton signals (δ 2.85 and 2.90 ppm) as presented in Figure 4. The difference of 0.05 ppm between the chemical shifts of protons at the C3 and C3′ positions in dimers **11a** and **11b** is comparable to the literature values described for analogous A/B ring systems. For example, the 3β-H signal in 3α-hydroxy-5α-androstane-17-one is located at 4.05 ppm [33], while the analogous signal (3α-H) in 3β-hydroxy-5β-androstane-17-one shows up at 4.11 ppm [34]. In addition, the chemical shifts of the C4 protons in dimer **11a** appear at δ 1.40 and 2.08 ppm, compared to 1.43 and 2.48 ppm for the corresponding proton signals in compound **11b**. The signal of the 4α proton in **11b** is significantly (0.40 ppm) shifted downfield compared to the analogous signal in **11a**. A similar difference (Δδ 0.43 ppm) between the chemical shifts of protons at C4 was reported in the literature for 5α- and 5β-androstan-17-ones [35]. The full assignment of signals in ^1^H and ^13^C NMR spectra was carried out based on different NMR techniques (DEPT, COSY, HMQC, and HMBC) and is presented in Appendix A (Appendix A).

A series of experiments were performed using mechanochemistry (Table 1; run 4), microwave (run 5), or ultrasound irradiation (run 6), but none of them produced reasonable yields of tosylate **4b** with hydroquinone reaction products.

In the next series of experiments, montmorillonite was used as a catalyst. Montmorillonite is a very soft phyllosilicate group of minerals that form when they precipitate from water solution as microscopic crystals, known as clay. It is named after Montmorillon in France. Montmorillonite, a member of the smectite group, is a 2:1 clay, meaning that it has two tetrahedral sheets of silica sandwiching a central octahedral sheet of alumina. The acid-treated montmorillonite clays have been extensively used in various catalytic processes [36]. Other heterogeneous Brønsted and Lewis acid catalysts could be developed using the cation-exchange ability of the montmorillonite interlayer. The organic syntheses using the above acid catalysts have several advantages: ease of preparing the solid catalysts, high catalytic activities, wide applicability to large molecules, simple workup procedure, and reusability of the catalyst [37]. Montmorillonite was also used for the preparation of ethers from alcohols and phenols. A comprehensive review of the catalytic *O*-alkylation of phenol and hydroquinone has recently appeared [38]. Also, the preparation of cholesteryl ethers from cholesterol and various alcohols or phenols has been described [39].

Before we started studying reactions catalyzed with montmorillonite, the reaction of androst-5-en-3β-ol-17-one tosylate (**4b**) with hydroquinone in the presence of basic Al_2_O_3_ was carried out. Unfortunately, the reaction was dirty and led to *i*-steroid products (e.g., 3α,5α-cyclo-androst-6-en-17-one) predominately. Therefore, a conclusion has been drawn that basic conditions should be avoided, and further reactions were carried out with K10 (commercial montmorillonite) as a catalyst under different conditions (Table 2). For these reactions, a non-activated androst-5-en-3β-ol-17-one (**4a**) was used instead of tosylate **4b**. In addition to the previously described products, new dimeric products, i.e., disteroidal ether **12** and the desired hydroquinone disteroidal diether **1,** appeared among the reaction products. The synthesis of disteroidal ethers was previously described. They can be efficiently prepared from Δ^5^-steroids by treating them with montmorillonite in a dichloromethane solution [40] or by an electrochemical method [41]. In both cases, a mesomeric homoallylic carbocation (Figure 1) is formed as an intermediate, which finally reacts with the starting sterol.

The reaction was optimized against the quantity of catalyst used. With 500 mg of K10 activated at 120 °C, a 10% yield of dimer **1** was achieved (Table 2; run 1); higher amounts of catalyst deteriorated (run 2) the reaction result. The temperature of the montmorillonite activation is very important. Montmorillonite changes its properties during activation due to the desorption of water, dehydration, and alteration of crystalline structure. The activation at 120 °C causes water desorption, which exposes acid groups. This increases the number and strength of acid centers. A higher temperature of activation results in partial dehydration of Si-OH and Al-OH groups. Then the number of Brønsted acid sites decreases, while the number of Lewis acid (e.g., Al^3+^) sites increases. The reactions of hydroquinone with 2.5 equiv. of DHEA (**4a**) in CHCl_3_ at reflux catalyzed with K10 (500 mg) activated at different temperatures were carried out (runs 3–6). The best result was obtained with K10 activated at 280 °C (run 5). The desired dimer **1** was obtained in 23% under optimized conditions. Further attempts to increase the yield of dimer **1** failed. The experiments presented in Table 2 were conducted in 5 mL of CHCl_3_ at reflux. The reaction carried out as in the case of run 5, but in a more diluted solution (10 mL of solvent), gave essentially the same result (23% of **1**). The reaction performed in 1,2-dichloroethane at reflux (83 °C) provided only 13% of **1**, while in other solvents at reflux (cyclohexane, toluene, xylenes, acetonitrile, acetone, THF, and dioxane), dimer **1** was not formed at all. Also, the attempts to carry out the reaction at room temperature were completely unsuccessful. In the next series of experiments, montmorillonite activated with hydrochloric acid was used. With this catalyst dried at 280 °C (H^+^-K10), up to a 57% yield of disteroidal ether **12** (Table 2; runs 7 and 8) could be achieved, but the yield of dimer **1** was negligible. It seems that increasing the number of Brønsted reactive sites deteriorates the yield of dimer **1**. Some experiments were carried out with montmorillonite activated with the ions Ti^4+^ or Cu^2+^. However, this modification of the catalyst promoted only the elimination processes (runs 9 and 10).

The optimal reaction conditions (as in run 5) were applied to analogous reactions of other sterols—diosgenin **5a** and cholesterol **6a**. In the latter case, the yield of desired dimer **3** was slightly better (28%) than that achieved for DHEA (**4a**). The dimer **3** was accompanied by dicholesteryl ether (22%), hydroquinone mono cholesteryl ether (10%), and the elimination products (35%). The yield of diosgenin-derived dimer **2** was lower (12%) under the same conditions. The corresponding disteroidal ether was isolated in tiny amounts [42].

In the next experiments, androst-5-en-3β-ol-17-one tosylate (**4b**) and cholesteryl tosylate (**6b**) served as substrates. The advantage of using tosylates instead of free sterols is that disteroidal ethers cannot be formed under these conditions, which is important from the point of view of reaction product separation. Also, dimers **11a**–**c** (or analogous compounds derived from cholesterol) were formed only in negligible amounts. In addition, much milder reaction conditions were used, resulting in fewer by-products. The results of tosylate reactions are shown in Table 3. It should be noted that all reactions occurred at ambient temperature and required less catalyst. In the case of the androstane series (runs 1–3), the best yield of the desired dimer **1** (31%) was observed for the reaction catalyzed by Ti^4+^ activated montmorillonite K10 (run 3). Without Ti^4+^ activation, the yield of **1** was 23%, provided that K10 was calcined at 500 °C (run 2). The reactions of cholesteryl tosylate **6b** with hydroquinone (runs 4–6) worked much better and provided dimer **3** in excellent yields. Especially the catalyst dried at 500 °C (run 5) was found to be very active in promoting the reaction, even in low quantities (50 mg). The likely explanation for the high activity of montmorillonite K10 against cholesterol **6a** and its tosylate **6b** is the lack of additional groups in these substrates capable of binding to the active catalyst sites.

## 3. Materials and Methods

### 3.1. General Experimental Data

All solvents were freshly distilled prior to use. Anhydrous solvents were prepared by distillation over appropriate drying agents under an argon atmosphere. The stabilizer (ethanol) contained in commercially available chloroform was removed before use. The reactions were monitored by TLC on silica gel plates 60 F254, and spots were visualized either by a UV hand lamp or by charring with molybdophosphoric acid/cerium(IV) sulfate in H_2_SO_4_. The reaction products were isolated by chromatographic methods using silica gel pore size 40 Å (70–230 mesh). ^1^H and ^13^C NMR (400 and 100 MHz, respectively) spectra of all compounds were recorded using a Bruker Avance II spectrometer in a CDCl_3_ or CDCl_3_/MeOD mixture and referenced to TMS (0.0 ppm) and CDCl_3_ (77.0 ppm), respectively. Infrared spectra were recorded using Attenuated Total Reflectance (ATR) as solid samples with a Nicolet 6700 FT-IR spectrometer. Mass spectra were obtained with an Accurate-Mass Q-TOFLC/MS 6530 spectrometer with electrospray ionization (ESI). Melting points were determined on a Kofler bench melting point apparatus. 

Tosylates **4b** [43] and **6b** [44] were prepared according to literature procedures.

Unmodified montmorillonite K10 or metal cation-exchanged montmorillonites: H^+^-K10, Ti^4+^-K10, and Cu^2+^-K10 were used as catalysts. Modified forms: H^+^-K10 [45], Ti^4+^-K10 [46], and Cu^2+^-K10 [47] were obtained according to the literature procedures. All catalysts were activated prior to use by heating at high temperatures (unmodified montmorillonite K10 at 120 °C, 200 °C, 280 °C, 400 °C, or 500 °C; H^+^-K10 at 280 °C; Ti^4+^-K10 at 280 °C or 500 °C; and Cu^2+^-K10 at 120 °C). Characterization data and NMR spectra are presented in the supporting information.

### 3.2. General Experimental Procedure for the Formation of All New Compounds

#### 3.2.1. Solvent-Free Reaction of Tosylate **4b** with Hydroquinone (Table 1, Run No. 2)

The mixture of tosylate **4b** (100 mg. 0.2 mmol) and hydroquinone (11 mg. 0.1 mmol) was stirred and heated to 120 °C for 5 min under an argon atmosphere. Then, the reaction mixture was cooled to room temperature and subjected to silica gel column chromatography, which resulted in the separation of compounds: **7** and **8** (eluted with hexane in 23% yield), **11a** (eluted with hexane/ethyl acetate 97:3 mixture in 7% yield), **11c** (eluted with hexane/ethyl acetate 95:5 mixture in 9% yield), **10a** (eluted with hexane/ethyl acetate 93:7 mixture in <5% yield), **11b** (eluted with hexane/ethyl acetate 93:7 mixture in 5% yield), **9** (eluted with hexane/ethyl acetate 87:13 mixture in 12% yield), and **10b** (eluted with hexane/ethyl acetate 87:13 mixture in <5% yield). 

**Compound 7:** white solid (hexane/CH_2_Cl_2_); mp 85–87 °C; Rf = 0.32 (hexane/ethyl acetate 9:1); IR (ATR) ν_max_ 2912, 2856, 1739 cm^–1^; ^1^H NMR (CDCl_3_, 400 MHz) δ 5.95 (1H, d, *J =* 9.9 Hz, H-4), 5.63 (1H, m, H-3), 5.42 (1H, m, H-6), 2.48 (1H, dd, *J =* 19.2 Hz, *J =* 8.8 Hz, H-16β), 0.99 (3H, s, H-19), 0.93 (3H, s, H-18); ^13^C NMR (CDCl_3_, 100 MHz) δ 221.1 (C), 141.6 (C), 128.7 (CH), 125.4 (CH), 122.1 (CH), 51.9 (CH), 48.5 (CH), 47.7 (C), 35.8 (CH_2_), 35.3 (C), 33.7 (CH_2_), 31.44 (CH_2_), 31.41 (CH), 30.6 (CH_2_), 23.0 (CH_2_), 21.8 (CH_2_), 20.3 (CH_2_), 18.8 (CH_3_), 13.7 (CH_3_); HRMS *m/z* 271.2054 (calcd for C_19_H_27_O^+^, 271.2056).

**Compound 8:** white solid; Rf = 0.36 (hexane/ethyl acetate 9:1); IR (ATR) ν_max_ 2916, 1736, 1506 cm^–1^; ^1^H NMR (CDCl_3_, 400 MHz) δ 5.99 (1H, dd, *J* = 9.8 Hz, *J* = 2.5 Hz, H-4), 5.56 (1H, d, *J* = 9.8 Hz, H-7), 5.47 (1H, m, H-3), 0.98 (3H, s, H-19), 0.96 (3H, s, H-18); ^13^C NMR (CDCl_3_, 100 MHz) δ 220.7 (C), 142.2 (C), 129.9 (CH), 125.6 (CH), 124.7 (CH), 51.6 (CH), 49.6 (CH), 48.3 (C), 36.5 (CH), 35.8 (CH_2_), 34.8 (C), 34.5 (CH_2_), 31.6 (CH_2_), 25.3 (CH_2_), 21.5 (CH_2_), 20.2 (CH_2_), 18.4 (CH_2_), 18.3 (CH_3_), 13.8 (CH_3_); HRMS *m/z* 271.2061 (calcd for C_19_H_27_O^+^, 271.2056).

**Compound 10a:** white solid; Rf = 0.66 (3 × hexane/ethyl acetate 74:26); IR (ATR) ν_max_ 3325, 2927, 1737, 1241, 1206, 1190, 1153, 1077, 1053, 813, 785 cm^–1^; ^1^H NMR (CDCl_3_, 400 MHz) δ 6.67 (1H, d, *J* = 8.6 Hz, H-Ar), 6.60 (1H, dd, *J* = 8.6 Hz, *J* = 3.0 Hz, H-Ar), 6.48 (1H, d, *J* = 3.0 Hz, H-Ar), 4.26 (1H, bs, -OH), 2.87 (1H, m, H-3β), 2.46 (1H, dd, *J* = 18.9 Hz, *J* = 8.3 Hz, H-16β), 1.05 (3H, s, H-19), 0.88 (3H, s, H-18); ^13^C NMR (CDCl_3_, 100 MHz) δ 221.4 (C), 150.1 (C), 147.9 (C), 128.0 (C), 115.3 (CH), 114.2 (CH), 113.9 (CH), 78.5 (C), 51.2 (CH), 47.8 (C), 46.2 (CH), 42.1 (C), 35.9 (CH_2_), 34.4 (CH), 33.6 (CH_2_), 32.7 (CH), 31.8 (CH_2_), 31.5 (CH_2_), 29.41 (CH_2_), 29.36 (CH_2_), 24.8 (CH_2_), 21.7 (CH_2_), 20.0 (CH_2_), 16.2 (CH_3_), 13.8 (CH_3_); HRMS *m/z* 381.2428 (calcd for C_25_H_33_O_3_^+^, 381.2424).

**Compound 11a:** white solid; Rf = 0.77 (3 × hexane/ethyl acetate 74:26); IR (ATR) ν_max_ 2915, 2853, 1737, 1242, 1205, 1151, 1001, 813, 785 cm^–1^; ^1^H NMR (CDCl_3_, 400 MHz) δ 6.41 (2H, s, H-Ar), 2.85 (2H, m, H-3β), 2.46 (2H, dd, *J* = 18.9 Hz, *J* = 8.2 Hz, H-16β), 1.04 (6H, s, H-19), 0.88 (6H, s, H-18); ^13^C NMR (CDCl_3_, 100 MHz) δ 221.5 (2 × C), 148.4 (2 × C), 126.1 (2 × C), 112.8 (2 × CH), 78.1 (2 × C), 51.2 (2 × CH), 47.8 (2 × C), 46.1 (2 × CH), 42.1 (2 × C), 35.9 (2 × CH_2_), 34.4 (2 × CH), 33.7 (2 × CH_2_), 32.6 (2 × CH), 32.2 (2 × CH_2_), 31.6 (2 × CH_2_), 29.5 (2 × CH_2_), 29.3 (2 × CH_2_), 24.8 (2 × CH_2_), 21.7 (2 × CH_2_), 20.0 (2 × CH_2_), 16.1 (2 × CH_3_), 13.8 (2 × CH_3_); HRMS *m/z* 651.4408 (calcd for C_44_H_59_O_4_^+^, 651.4408).

**Compound 11b:** white solid; Rf = 0.66 (3 × hexane/ethyl acetate 74:26); IR (ATR) ν_max_ 2914, 2853, 1737, 1242, 1205, 813, 785 cm^–1^; ^1^H NMR (CDCl_3_, 400 MHz) δ 6.43 (2H, s, H-Ar), 2.90 (2H, m, H-3α), 2.51–2.43 (4H, m, H-4α and H-16β), 1.03 (6H, s, H-19), 0.88 (6H, s, H-18); ^13^C NMR (CDCl_3_, 100 MHz) δ 221.0 (2 × C), 149.1 (2 × C), 125.4 (2 × C), 113.1 (2 × CH), 78.4 (2 × C), 51.6 (2 × CH), 47.9 (2 × C), 43.8 (2 × CH), 42.9 (2 × C), 35.9 (2 × CH_2_), 34.6 (2 × CH), 34.4 (2 × CH_2_), 32.6 (2 × CH), 31.6 (2 × CH_2_), 30.8 (2 × CH_2_), 28.7 (2 × CH_2_), 27.9 (2 × CH_2_), 27.1 (2 × CH_2_), 21.8 (2 × CH_2_), 20.6 (2 × CH_2_), 17.6 (2 × CH_3_), 13.8 (2 × CH_3_); HRMS *m/z* 651.4401 (calcd for C_44_H_59_O_4_^+^, 651.4408).

**Compound 11c:** white solid; Rf = 0.71 (3 × hexane:ethyl acetate 74/26); IR (ATR) ν_max_ 2929, 2870, 1737, 1241, 1194, 1152, 785 cm^–1^; ^1^H NMR (CDCl_3_, 400 MHz) δ 6.43 (2H, s, H-Ar), 2.91 (1H, m, H-3α), 2.85 (1H, m, H-3β), 1.03 (3H, s, H-19), 1.01 (3H, s, H-19′), 0.88 (s, 6H, H-18 and H-18′); ^13^C NMR (CDCl_3_, 100 MHz) δ 221.4 (C), 220.9 (C), 149.0 (C), 148.7 (C), 126.1 (C), 125.2 (C), 113.0 (CH), 112.7 (CH), 78.3 (C), 78.1 (C), 51.6 (CH), 51.2 (CH), 47.8 (2 × C), 46.1 (CH), 43.9 (CH), 42.9 (C), 42.1 (C), 35.9 (2 × CH_2_), 34.5 (CH), 34.4 (CH), 34.4 (CH_2_), 33.6 (CH_2_), 32.5 (2 × CH), 32.0 (CH_2_), 31.6 (CH_2_), 31.5 (CH_2_), 30.7 (CH_2_), 29.5 (CH_2_), 29.4 (CH_2_), 28.8 (CH_2_), 27.9 (CH_2_), 27.0 (CH_2_), 24.8 (CH_2_), 21.8 (CH_2_), 21.7 (CH_2_), 20.6 (CH_2_), 20.0 (CH_2_), 17.6 (CH_3_), 16.2 (CH_3_), 13.8 (2 × CH_3_); HRMS *m/z* 651.4403 (calcd for C_44_H_59_O_4_^+^, 651.4408).

#### 3.2.2. Optimal Procedure for Preparation of Dimer **1** in the Montmorillonite K10 Catalyzed Reaction of Androst-5-en-3β-ol-17-one (**4a**) with Hydroquinone (Table 2, Run No. 5)

A stirred mixture of androst-5-en-3β-ol-17-one (**4a**) (100 mg, 0.35 mmol), hydroquinone (15.4 mg, 0.14 mmol), and unmodified montmorillonite K10 activated at 280 °C (500 mg) in dry chloroform (5 mL) was gently refluxed under argon, and the reaction progress was monitored by TLC. After completion of the reaction (4 h), the suspension was filtered through a sintered glass funnel, and the precipitate was washed with a methanol/chloroform 2:8 mixture (3 × 50 mL). The filtrate was evaporated in a vacuum. The residue was subjected to column chromatography on silica gel, which resulted in the separation of compounds **7** and **8** (eluted with hexane in 28% yield), **12** (eluted with hexane/ethyl acetate 95:5 mixture in 7% yield), **1** (eluted with hexane/ethyl acetate 94:6 mixture in 23% yield), **9** (eluted with hexane/ethyl acetate 87:13 mixture in 12% yield), and small amounts of compounds **11a**, **11b,** and **11c** (total <5%).
**Compound 1:** colorless crystals (CH_2_Cl_2_/ethyl acetate); mp 252–254 °C; Rf = 0.50 (3 × benzene/ethyl acetate 94:6); IR (ATR) ν_max_ 2938, 2907, 1744, 1731, 1501, 1214, 1043, 1029, 815 cm^–1^; ^1^H NMR (CDCl_3_, 400 MHz) δ 6.82 (4H, s, H-Ar), 5.42 (2H, m, H-6), 4.00 (2H, m, H-3α), 1.09 (6H, s, H-19), 0.91 (6H, s, H-18); ^13^C NMR (CDCl_3_, 100 MHz) δ 221.0 (2 × C), 151.8 (2 × C), 140.7 (2 × C), 121.4 (2 × CH), 117.4 (4 × CH), 77.9 (2 × CH), 51.8 (2 × CH), 50.3 (2 × CH), 47.5 (2 × C), 38.8 (2 × CH_2_), 37.1 (2 × CH_2_), 37.0 (2 × C), 35.8 (2 × CH_2_), 31.5 (2 × CH), 31.4 (2 × CH_2_), 30.8 (2 × CH_2_), 28.3 (2 × CH_2_), 21.9 (2 × CH_2_), 20.4 (2 × CH_2_), 19.4 (2 × CH_3_), 13.5 (2 × CH_3_); HRMS *m/z* 651.4409 (calcd for C_44_H_59_O_4_^+^, 651.4408).
**Compound 9:** colorless crystals (CH_2_Cl_2_/ethyl acetate); mp 277–279 °C; Rf = 0.48 (3 × hexane/ethyl acetate 74:26); IR (ATR) ν_max_ 3311, 2948, 2864, 1710, 1505, 1211, 1028, 819 cm^−1^; ^1^H NMR (CDCl_3_/MeOD, 400 MHz) δ 6.76 (2H, d, *J =* 19.2 Hz, H-Ar), 6.71 (2H, d, *J* = 19.1 Hz, H-Ar), 5.37 (1H, m, H-6), 3.93 (1H, m, H-3α), 1.05 (3H, s, H-19), 0.87 (3H, s, H-18); ^13^C NMR (CDCl_3_/MeOD, 100 MHz) δ 221.9 (C), 150.8 (C), 150.7 (C), 140.7 (C), 121.3 (CH), 117.9 (2 × CH), 115.8 (2 × CH), 78.3 (CH), 51.7 (CH), 50.2 (CH), 47.6 (C), 38.8 (CH_2_), 37.0 (CH_2_), 36.9 (C), 35.8 (CH_2_), 31.4 (CH), 31.3 (CH_2_), 30.7 (CH_2_), 28.2 (CH_2_), 21.8 (CH_2_), 20.3 (CH_2_), 19.4 (CH_3_), 13.5 (CH_3_); HRMS *m/z* 381.2438 (calcd for C_25_H_33_O_3_^+^, 381.2424).
**Compound 12:** colorless crystals (hexane/CH_2_Cl_2_); mp 268–269 °C; Rf = 0.45 (3 × benzene/ethyl acetate 94:6); IR (ATR) ν_max_ 2931, 2895, 1731, 1094, 1058, 1005 cm^–1^; ^1^H NMR (CDCl_3_, 400 MHz) δ 5.38 (2H, m, H-6), 3.30 (2H, m, H-3α), 2.47 (2H, dd, *J* = 19.2 Hz, *J* = 8.6 Hz, H-16β), 1.04 (6H, s, H-19), 0.89 (6H, s, H-18); ^13^C NMR (CDCl_3_, 100 MHz) δ 221.2 (2 × C), 141.5 (2 × C), 120.6 (2 × CH), 76.2 (2 × CH), 51.8 (2 × CH), 50.3 (2 × CH), 47.5 (2 × C), 40.0 (2 × CH_2_), 37.3 (2 × CH_2_), 37.0 (2 × C), 35.8 (2 × CH_2_), 31.5 (2 × CH), 31.4 (2 × CH_2_), 30.8 (2 × CH_2_), 29.3 (2 × CH_2_), 21.9 (2 × CH_2_), 20.3 (2 × CH_2_), 19.4 (2 × CH_3_), 13.5 (2 × CH_3_).


Analogously, steroid dimers **2** (12%) and **3** (28%) were obtained from diosgenin (**5a**) and cholesterol (**6a**), respectively, according to the procedure described above for androst-5-en-3β-ol-17-one (**4a**). The elimination products (~35% in both cases), disteroidal ethers (12% and 22%, respectively), and hydroquinone mono steroidal ethers (12% and 10%, respectively) were also formed.
**Compound 2:** colorless crystals (hexane/CH_2_Cl_2_); mp 307–309 °C; Rf = 0.38 (3 × benzene/ethyl acetate 94:6); IR (ATR) ν_max_ 2925, 1502, 1225, 1050, 1016, 809 cm^–1^; ^1^H NMR (CDCl_3_, 400 MHz) δ 6.82 (4H, s, H-Ar), 5.38 (2H, m, H-6), 4.43 (2H, m, H-16), 3.98 (2H, m, H-3α), 3.49 (2H, m, H-26β), 3.39 (2H, t, *J* = 10.9 Hz, H-26α), 1.08 (6H, s, H-19), 0.99 (6H, d, *J* = 6.9 Hz, H-21), 0.81 (6H, s, H-18), 0.80 (6H, d, *J* = 5.0 Hz, H-27); ^13^C NMR (CDCl_3_, 100 MHz) δ 151.8 (2 × C), 140.5 (2 × C), 121.9 (2 × CH), 117.4 (4 × CH), 109.3 (2 × C), 80.8 (2 × CH), 78.0 (2 × CH), 66.8 (2 × CH_2_), 62.1 (2 × CH), 56.5 (2 × CH), 50.1 (2 × CH), 41.6 (2 × CH), 40.3 (2 × C), 39.8 (2 × CH_2_), 38.8 (2 × CH_2_), 37.2 (2 × CH_2_), 37.0 (2 × C), 32.1 (2 × CH_2_), 31.8 (2 × CH_2_), 31.43 (2 × CH), 31.38 (2 × CH_2_), 30.3 (2 × CH), 28.8 (2 × CH_2_), 28.3 (2 × CH_2_), 20.8 (2 × CH_2_), 19.4 (2 × CH_3_), 17.1 (2 × CH_3_), 16.3 (2 × CH_3_), 14.5 (2 × CH_3_); HRMS *m/z* 903.6502 (calcd for C_60_H_87_O_6_^+^, 903.6497).
**Diosgenin derived hydroquinone mono steroidal ether:** colorless crystals (hexane/CH_2_Cl_2_); mp 165–166 °C; Rf = 0.35 (hexane/ethyl acetate 88:12); IR (ATR) ν_max_ 3309, 2929, 1507, 1210, 1046, 1012, 830 cm^–1^; ^1^H NMR (CDCl_3_, 400 MHz) δ 6.80 (2H, d, *J* = 9.0 Hz, H-Ar), 6.75 (2H, d, *J* = 9.0 Hz, H-Ar), 5.38 (1H, m, H-6), 4.76 (1H, bs, -OH), 4.43 (1H, m, H-16), 3.97 (1H, m, H-3α), 3.49 (1H, m, H-26β), 3.39 (1H, t, *J* = 10.9 Hz, H-26α), 1.07 (3H, s, H-19), 0.99 (3H, d, *J* = 6.9 Hz, H-21), 0.81 (3H, s, H-18), 0.80 (3H, d, *J* = 6.4 Hz, H-27); ^13^C NMR (CDCl_3_, 100 MHz) δ 151.6 (C), 149.8 (C), 140.5 (C), 121.9 (CH), 117.8 (CH), 116.0 (CH), 109.4 (C), 80.8 (CH), 78.3 (CH), 66.9 (CH_2_), 62.1 (CH), 56.5 (CH), 50.1 (CH), 41.6 (CH), 40.3 (C), 39.8 (CH_2_), 38.8 (CH_2_), 37.1 (CH_2_), 37.0 (C), 32.1 (CH_2_), 31.8 (CH_2_), 31.42 (CH), 31.37 (CH_2_), 30.3 (CH), 28.8 (CH_2_), 28.3 (CH_2_), 20.9 (CH_2_), 19.4 (CH_3_), 17.1 (CH_3_), 16.3 (CH_3_), 14.5 (CH_3_); HRMS *m/z* 507.3476 (calcd for C_33_H_47_O_4_^+^, 507.3469).

#### 3.2.3. Optimal Procedure for Preparation of Dimer **12** in the Montmorillonite K10 Catalyzed Reaction of Androst-5-en-3β-ol-17-one (**4a**) (Table 2, Run No. 8)

A stirred mixture of androst-5-en-3β-ol-17-one (**4a**) (100 mg, 0.35 mmol), hydroquinone (15.4 mg, 0.14 mmol), and modified montmorillonite H^+^-K10 activated at 280 °C (500 mg) in dry chloroform (5 mL) was heated to 50 °C for 24 h under argon, and the reaction progress was monitored by TLC. After completion of the reaction, the suspension was filtered through a sintered glass funnel, and the precipitate was washed with a methanol/chloroform 2:8 mixture (3 × 50 mL). The filtrate was evaporated in a vacuum. The residue was subjected to silica gel column chromatography, which resulted in separation of compounds **7** and **8** (eluted with hexane in 13% yield), **12** (eluted with hexane/ethyl acetate 95:5 mixture in 57% yield), **1** (eluted with hexane/ethyl acetate 94:6 mixture in 1% yield), and **9** (eluted with hexane/ethyl acetate 87:13 mixture in 7% yield).

#### 3.2.4. Optimal Procedure for Preparation of Dimer **1** in the Montmorillonite K10 Catalyzed Reaction of Tosylate **4b** with Hydroquinone (Table 3, Run No. 3)

A mixture of tosylate **4b** (88.5 mg; 0.2 mmol), hydroquinone (8.4 mg; 0.076 mmol), and modified montmorillonite Ti^4+^-K10 activated at 280 °C (300 mg) in dry chloroform (5 mL) was stirred under argon, and the reaction progress was monitored by TLC. After completion of the reaction (3 days), the suspension was filtered through a sintered glass funnel, and the precipitate was washed with a methanol/chloroform 2:8 mixture (3 × 50 mL). The filtrate was evaporated in a vacuum. The residue was subjected to column chromatography on silica gel, which resulted in the separation of compounds **7** and **8** (eluted with hexane in 28% yield), **1** (eluted with hexane/ethyl acetate 94:6 mixture in 31% yield), and **9** (eluted with hexane/ethyl acetate 87:13 mixture in 19% yield).

The reaction of tosylate **6b** with hydroquinone was carried out according to the procedure described above for tosylate **4b,** but using montmorillonite K10 activated at 500 °C (50 mg) as a catalyst (Table 3, Run No. 5). Dimer **3** was obtained with a 67% yield. Small amounts of elimination products (10%) were also formed.

## 4. Conclusions

The reactions of sterols and their tosylates with hydroquinone were studied. They provided different steroid dimers depending on the reaction conditions. The solvolytic reactions of DHEA tosylate afforded the elimination products in addition to three stereoisomeric steroid dimers **11a**–**11c**, which resulted from the nucleophilic attack of the C2 carbon atom of hydroquinone on the C3 position of the steroid, followed by an intramolecular addition to the C5–C6 double bond. The cause of reactions catalyzed by montmorillonite was different. The major reaction products were the steroid dimers with a 3,3′-*O*,*O*-1,4-phenylene linker **1**–**3** and the disteroidal ethers **12**. The formation of the latter compounds was suppressed by using sterol tosylates for the montmorillonite-catalyzed reactions with hydroquinone. As a result, an excellent yield (67%) of hydroquinone dicholesteryl diether (dimer **3**) was achieved.

## Data Availability

All generated data will be available from the authors upon request.

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
