# Peer review of "Montmorillonite Catalyzed Synthesis of Novel Steroid Dimers"

_molecules, 2023, doi:10.3390/molecules28207068_

Round 1

Reviewer 1 Report

The manuscript deals with the dimerization reaction of three steroids, under different conditions (solvent, temperature, and catalyst) and the authors explain the effect of the main variations.

The manuscript is suitable for publication on “Molecules” after a minor revision, according to the following:

Comments and suggestions:

-Page 3, line 58 The authors should add a brief sentence about the molecular rotors.

-             line 64 (25R)- spirost…. Instead of 25R-spirost….

- Scheme 1 Add square brackets around the mesomeric cations since they are not isolated reaction intermediates.

                   On the arrows before compound 7 add hydroquinone (name or formula).

Table 1, run 1 The reaction time is 24h; at page 3, line 77 the reaction time is 3 days.

Tables 1,2, and 3 The entries number not in bold; the bold character is used to indicate number of the compounds.

Table 3 The authors should indicate that entries 1-3 are referred to compound 4b and entries 4-6 to compound 6b.

In Materials and Methods section, to make easier the reading, the Tables entry number of each preparation should be inserted.

Author Response

Response to Reviewer 1 comments

The manuscript deals with the dimerization reaction of three steroids, under different conditions (solvent, temperature, and catalyst) and the authors explain the effect of the main variations.

The manuscript is suitable for publication on “Molecules” after a minor revision, according to the following:

Comments and suggestions:

Page 3, line 58 The authors should add a brief sentence about the molecular rotors.

A brief information about the molecular rotors has been added on page 3, lines 59-62.

-             line 64 (25R)- spirost…. Instead of 25R-spirost….

The suggested change has been made.

- Scheme 1 Add square brackets around the mesomeric cations since they are not isolated reaction intermediates.

On the arrows before compound 7 add hydroquinone (name or formula).

Both suggested changes in Scheme 1 have been made.

Table 1, run 1 The reaction time is 24h; at page 3, line 77 the reaction time is 3 days.

We apologize for this inaccuracy. The actual reaction time was 24 h. Error on page 3 has been corrected.

Tables 1, 2, and 3 The entries number not in bold; the bold character is used to indicate number of the compounds.

The suggested changes have been made.

Table 3 The authors should indicate that entries 1-3 are referred to compound 4b and entries 4-6 to compound 6b.

Though this information is given in the legend of Table 3, the compound numbers have been added to the table.

In Materials and Methods section, to make easier the reading, the Tables entry number of each preparation should be inserted.

The entry numbers of each preparation have been added to the Materials and Methods section.

The authors thank the reviewer for his valuable comments.

Reviewer 2 Report

Morzycki and co-workers developed a steroid dimer synthesis strategy by using montmorillonite catalyzed condition. Without montmorillonite as catalyst, the desired dimer product is barely produced with several side products generated. The authors also discussed the possible side products generation mechanism. The use of montmorillonite helped to improve the selectivity of the reaction towards the desired steroid dimer. Overall, the manuscript is well-written and well-organized. The results are supported by a good amount of data. This work can be potentially useful. I recommend its publication in Molecules after addressing the following aspects.

1.       In Table 3, the authors showed the reaction of 4b and 6b with hydroquinone catalyzed with K10, have the authors also tested with 5b? If it is, the authors should also include the data in the table.

2.       In supporting information, the characterization of compound 3 was missing.

3.       The authors tested tosylated compounds for the reaction, have the authors tested with other leaving groups?

4.       In both Table 2 and Table 3, the authors used different amounts of K10 catalyst, what is the rationale behind this? Have the authors done a systematic study of the amount of montmorillonite to the reactivity of the reaction?

Author Response

Response to Reviewer 2 comments

Morzycki and co-workers developed a steroid dimer synthesis strategy by using montmorillonite catalyzed condition. Without montmorillonite as catalyst, the desired dimer product is barely produced with several side products generated. The authors also discussed the possible side products generation mechanism. The use of montmorillonite helped to improve the selectivity of the reaction towards the desired steroid dimer. Overall, the manuscript is well-written and well-organized. The results are supported by a good amount of data. This work can be potentially useful. I recommend its publication in Molecules after addressing the following aspects.

  1. In Table 3, the authors showed the reaction of 4b and 6b with hydroquinone catalyzed with K10, have the authors also tested with 5b? If it is, the authors should also include the data in the table.

Sorry, but we did not test the reaction with 5b. This was because the previous experiments with free sterols were disappointing for diosgenin (a much lower yield of the desired dimer was obtained than in for DHEA or cholesterol). We assume that this may be due to the strong adsorption of spirostane on the catalyst, causing its deactivation.

  1. In supporting information, the characterization of compound 3 was missing

Compound 3 turned out to be identical with the product of an electrochemical reaction, which we had previously obtained with low yield. Since full characterization of this compound has been described in ref. 32: (Tomkiel, A.M.; Kowalski, J.; Płoszyńska, J.; Siergiejczyk, L.; Łotowski, Z.; Sobkowiak, A.; Morzycki, J.W. Electrochemical synthesis of glycoconjugates from activated sterol derivatives, Steroids 2014, 82, 60-67; DOI: 10.1016/j.steroids.2014.01.007), we didn't think there is a need to include this description in the Supplementary Information.

However, the complete characterization of compound 3 has been included in the revised version of Supporting Information.

  1. The authors tested tosylated compounds for the reaction, have the authors tested with other leaving groups?

Other leaving groups were not tested.

  1. In both Table 2 and Table 3, the authors used different amounts of K10 catalyst, what is the rationale behind this? Have the authors done a systematic study of the amount of montmorillonite to the reactivity of the reaction?

For all substrates a systematic study have been done for each K-10 catalyst modification to find the optimal catalyst amount. The tables show only the results of experiments carried out under optimized conditions. The differences in the amount of catalyst needed for individual substrates are probably due to their structure. Cholesterol 6a (or its tosylate 6b) has no additional functional groups capable of binding to the active sites of the catalyst, therefore less catalyst is required than for DHEA 4a or its tosylate 4b.

The authors thank the reviewer for his valuable comments.